# Influence of Loading Directions on Dynamic Compressive Properties of Mill-Annealed Ti-6Al-4V Thick Plate

**DOI:** 10.3390/ma15207047

**Published:** 2022-10-11

**Authors:** Dongyang Qin, Shenglu Lu, Yulong Li

**Affiliations:** 1School of Aeronautics, Northwestern Polytechnical University, Xi’an 710072, China; 2Center for Additive Manufacturing, Royal Melbourne Institute of Technology, Melbourne, VIC 3000, Australia; 3School of Civil Aviation, Northwestern Polytechnical University, Xi’an 710072, China

**Keywords:** titanium alloys, twinning, texture, FIB, Hopkinson bar

## Abstract

This paper investigates the influence of loading directions on mechanical performance, damage behavior and failure mechanisms of a mill-annealed Ti-6Al-4V (TC4) alloy thick plate at the strain rate of 2000/s. The plate possesses {11-20} texture and consists of globular α grain, fine equiaxial α grain, α laminate that is parallel to the normal direction (ND) of the plate and grain-boundary β laminate. The yield strength and the flow stress of the plate are not affected by the loading directions, while the fracture strain in ND is 38.2% and 32.2% higher than that in the rolling direction (RD) and the traverse direction (TD). As it is loaded in the RD and TD, the deformation mechanism of the alloy is dislocation slip. However, the deformation mechanisms in ND are dislocation slip and {10-12}<10-1-1> twinning. The activation of {10-12}<10-1-1> twinning could delay the formation of the adiabatic shearing band (ASB). Multiple adiabatic shearing bands (ASBs) form as the compression direction is in the RD and TD. In contrast, as the compression direction is in ND, only one ASB could be observed. The dramatic adiabatic shear could not result in the dynamic recrystallization of the mill-annealed TC4 alloy but could lead to the formation of nano-sized α laminate. The compressive fracture mechanism of the alloy plate is the crack propagation in the main ASB, which is not affected by the loading directions. Here we attribute the superior dynamic failure strain in the ND of the plate to the {10-12}<10-1-1> twinning induced by {11-20}_α_ texture, cooperative deformation ability of the α laminate and higher shear strain within the ASB. The findings of our work are instructive for reducing foreign object damage to mill-annealed TC4 alloy fan blades.

## 1. Introduction

Owing to the excellent strength-to-weight ratio, high dynamic flow stress and prominent thermo-mechanical processing ability, Ti-6Al-4V (TC4) titanium alloy has been extensively used to fabricate the fan blades of aero engines [1]. During the take-off and landing of aircraft, the fan blades suffer from impact damage from foreign objects such as gravel, bolts and rivets [2]. Since foreign object damage (FOD) has a great impact on the fatigue properties of TC4 alloy blades, it is necessary to improve the anti-FOD capacity of the blades [3]. Although the FOD of the blades might be influenced by the impact speed, the impact angle, the mass of the foreign object, etc., the anti-FOD performance of the fan blades strongly depends on the dynamic mechanical properties of the parent TC4 alloy [3].

The dynamic mechanical performance of TC4 alloy has been extensively investigated. The well-known factors that might influence the dynamic mechanical performances of TC4 alloy include microstructure, deformation temperature and strain rate. The microstructure of the alloy could be classified into equiaxial, bimodal and lamellar, which is based on the morphology of α phase [4]. Equiaxial alloy has the lowest strength and the highest ductility; lamellar alloy demonstrates the highest strength and the lowest ductility; bimodal alloy exhibits the best combination of dynamic strength and plasticity [5,6,7]. Yield strength and flow stress of TC4 alloy are inversely proportional to deformation temperature during high-strain-rate deformation. Nasser et al. reported that the yield strength of equiaxial alloy decreases from 1760 MPa to 400 MPa as the temperature increases from 77 K to 1373 K [8]. Within the strain-rate range of 1000/s~8000/s, the yield strength and flow stress of TC4 alloy slightly increase with the strain rate [9,10,11]. Aside from the factors mentioned above, the dynamic properties of TC4 alloy are also influenced by texture. Gu et al. found that an equiaxial Ti-64 alloy rod exhibits obvious plasticity anisotropy during dynamic compression [12]. The fracture strain in the axial direction of the rod is as high as 0.4, while the fracture strain in the radial direction of the rod is only 0.17 [12].

It should be noted that the TC4 alloy plate with mill-annealed microstructure is the critical raw material of the fan blades. However, few researchers have studied the dynamic mechanical performance and the related dynamic mechanical property anisotropy of the mill-annealed TC4 alloy plate. In this study, dynamic compression testing on the RD, TD and ND of the mill-annealed TC4 alloy thick plate was conducted on the split Hopkinson pressure bar at the strain rate of 2000/s. This study aims to understand the effects of loading direction on the dynamic mechanical properties, dynamic deformation mechanism and adiabatic shearing behavior of the TC4 alloy plate. The results of the work indicate that the anti-FOD performance of a mill-annealed TC4 fan blade might be improved by the texture of the fan blade.

## 2. Material and Methods

### 2.1. Initial Microstructure of TC4 Alloy Plate

The mill-annealed TC4 alloy plate was supplied by Aero Engine Corporation of China. The normal chemical composition of the alloy was Ti-6Al-4V (in wt.%). The dimensions of the RD, TD and ND of the plate were 600 mm, 1200 mm and 25 mm, respectively. The microstructure of the RD–TD plane, RD–ND plane and TD–ND plane was characterized using a Zeiss Axiovert-A1 optical microscope. The texture for the α phase of the plate was investigated using an HKL electron backscattering diffraction (EBSD) detector coupled with a Zeiss Supra-55 scanning electron microscope (SEM). The pole figure of the alloy was plotted using the Channel 5 package.

### 2.2. Dynamic Compression Test

A specimen for dynamic compression with the dimensions of Φ5 mm × 5 mm was machined from the plate. The height directions of the RD specimen, TD specimen and ND specimen were parallel to the RD, TD and ND of the plate. The dynamic compression test was conducted on a Φ19 mm split Hopkinson pressure bar (SHPB). The striking bar, the incident bar and the transmitted bar were 450 mm, 1800 mm and 1800 mm. The resistance strain gages (1 KΩ, 0°) were glued on the bars to measure the strain pulse. The reflected strain (*ε_R_*(*t*)) of the incident bar and the transmitted strain pulse (*ε_T_*(*t*)) of the transmitted bar were recorded by a Nicolet Odyssey-XE data collecting system. By using the technique of single-pulse loading, the specimens were loaded to failure at the strain rate of 2000/s. The strain (*ε*(*t*)) and the stress (*σ*(*t*)) of the specimen were calculated by:$$\sigma \left(t\right)=E\left(\frac{A_{b}}{A_{s}}\right)\varepsilon_{T}\left(t\right)$$
$$\varepsilon \left(t\right)=\left(\frac{2C_{0}}{l_{s}}\right)\int_{0}^{t}\varepsilon_{R}\left(t\right)dt$$
where *E* is the elastic modulus of the steel bars, *A_b_* and *A_s_* are the area of the specimen and the area of the transmitted bar, *ε_T_*(*t*) is the transmitted strain pulse, *ε_R_*(*t*) is the reflected strain pulse, *C*_0_ is the elastic wave velocity for the steel bars and *l_s_* is the height of dynamic compression specimen.

### 2.3. Strain-Limited Dynamic Compression Test

The strain-limited dynamic compression test of RD, TD and ND specimens was conducted on the Φ19 mm SHPB by using the stopping ring. The strain rate was also 2000/s. The outer diameter and the inner diameter of the stopping ring were 19 mm and 12 mm. The design height of the stopping ring was 4.5 mm to achieve the compression strain of 0.1. Before the dynamic compression test, the stopping ring was fixed on the end of the transmitted bar using 3M-DP460 epoxy adhesive. Although the height of the stopping ring was 4.5 mm, the actual height of the strain-limited specimen after the dynamic compression was only 4.3 mm because of the elastic deformation of the stopping ring and the peeling of the epoxy adhesive layer.

### 2.4. Microstructure Characterization of Strain-Limited Specimen

The strain-limited specimens were mounted into resin and prepared into metallographic specimens. The normal of the observed surface for the metallographic specimens was parallel to the loading direction of the strain-limited specimens. The optical micrograph of the strain-limited specimens was recorded using a Zeiss Stemi2000c optical microscope. HKL-EBSD equipped in a Zeiss Supra-55 SEM and an FEI Themis Z probe-corrected transmission electron microscope (TEM) were used to investigate the dynamic deformation mechanism of the strain-limited specimens. The normal of the TEM foil was parallel to the loading direction.

### 2.5. Microstructure Characterization of ASB

Ruptured specimens were cut in half along the compression, mounted into resin and finally prepared into metallographic specimens. The microstructure of the ASB was observed using a Zeiss Stemi2000c optical microscope. FEI Helios 600 focused ion beam (FIB) SEM was used to machine the TEM specimen from the ASB. The dimension of the rectangle TEM foil was approximate 10 μm in width and 20 μm in length, and the normal of the TEM foil was parallel to the shearing direction. An FEI Themis Z probe-corrected TEM was used to investigate the microstructure of the ASB.

## 3. Results

### 3.1. Initial Microstructure of TC4 Plate

Figure 1a displays the optical microscopy photographs for the RD–TD plane, RD–ND plane and TD–ND plane. The α phase exhibits the bright contrast, while the β phase demonstrates the dark contrast. The globular α grains could be clearly seen in all directions. However, α laminate could only be observed in RD–ND plane and TD–ND plane, suggesting that the normal of α laminate is perpendicular to the ND. The β phase is distributed on the grain boundary of the α phase. Figure 1b illustrates the orientation imaging microscopy (OIM) micrograph for the α phase in the RD–TD plane. The diameter for the coarse α grain is approximately 25 μm, and the diameter for the fine α grain is less than 10 μm. The optical microscopy micrographs and OIM micrograph suggest that the alloy consists of coarse globular α grain, fine globular α grain, α laminate and β laminate.

Figure 1c presents the {0001}_α_, {11-20}_α_ and {10-10}_α_ pole figure (PF) of Figure 1b. In the {0001} PF, the maximum intensity region is located on the outer circle, and the separation angle between the pole and the RD is approximate 60°. These results indicate that the c-axis of most of the α grain is perpendicular to the normal of the RD–TD plane and that the separation angle between the c-axis and the RD is 60°. In the {11-20} PF, there is one pole that is located in the center of the PF, suggesting that the normal of {11-20} is approximately parallel to ND. In the {10-10} PF, there is one pole with a polar of 30° to the center of the PF. Since the separation angle between {10-10} and {11-20} is 30°, the results of the {10-10} PF also indicate that the normal of {11-20} is parallel to the ND, which is in agreement with the results of the {11-20} PF. Consequently, we conclude that the mill-annealed TC4 plate possesses the strong {11-20} texture.

### 3.2. Dynamic Compressive Mechanical Properties of TC4 Plate

Figure 2 shows stress–strain curves of the RD, TD and ND specimens deformed at the strain rate of 2000/s. In the plastic deformation region, the flow stress increases with the compressive strain, indicating that all the specimens exhibit strain-hardening behavior during dynamic compression. Table 1 summarizes the dynamic mechanical performance of the RD, TD and ND specimens. The dynamic yield strength of the ND specimen is 1.3% lower than that of the RD specimen and 3.1% lower than that of the TD specimen, indicating that the dynamic strength of the plate is almost isotropic. However, the dynamic fracture strain of the ND specimen is 38.2% higher than that of the RD specimen and 32.2% higher than that of the TD specimen, suggesting the plate is plastically anisotropic in dynamic compression.

### 3.3. Effects of Loading Direction on Deformation Mechanism of Strain-Limited Specimen

#### 3.3.1. Twinning of Strain-Limited Specimen

Figure 3a,b show the optical micrographs of the strain-limited RD specimen (ε = 10%). Coarse α grain, α laminate and β laminate could be still distinguished. Compared with the initial microstructure of the TD–ND plane, we find that the morphology of the α phase and the distribution of β laminate do not change evidently. In addition, the dark stripes form in a few globular α grains. Figure 3c,d display the optical micrographs of the strain-limited TD specimen (ε = 10%). Generally speaking, the microstructure of the TD specimen is almost the same as that of the RD–ND plane of the initial alloy. In addition, the dark stripes could also be observed in a few globular α grains. Figure 3e,f demonstrate the optical micrographs of the strain-limited ND specimen (ε = 10%). It is obvious that the volume fraction of the dark stripes is much higher: The dark stripes could be observed in most of the globular α grains.

Figure 4a shows the OIM micrograph of the strain-limited RD specimen. The dark stripe observed by optical microscopy is also found in the micrograph, which is marked by the white rectangles. Figure 4b demonstrates the misorientation of the line in region 1 and region 2. In region 1, the misorientation between the globular α grain (green area) and the stripe (yellow area) is 35°, suggesting that {11-21}<1-100> twinning is activated in the globular α grain. In region 2 the misorientation between the globular α grain (green area) and the stripe (red area) is 85°, indicating that the {10-12}<10-1-1> twinning forms. The observation of {10-12}<10-1-1> twinning is in agreement with the work of Coghe et al. and the work of Wielewski et al., in which {10-12}<10-1-1> twinning was also found during the high-strain-rate deformation of TC4 alloy [13,14]. Figure 4c displays the OIM micrograph of the strain-limited TD specimen. The misorientation profile of the line in the rectangle region is shown in Figure 4d. The misorientation between the globular α grain and the stripe is 85°, indicating that {10-12}<10-1-1> twinning is also activated. It should be noted that the {11-21}<1-100> twinning could not be observed in the strain-limited TD specimen. Although the twinning of the strain-limited TD specimen is different from that of the strain-limited RD specimen, the total volume fraction of {10-12}<10-1-1> twinning is extremely low. Figure 4e shows the OIM micrograph of the strain-limited ND specimen. The misorientation of the line in the rectangle region is shown in Figure 4f. The misorientation between the globular α grain and the stripe is 85°, indicating that the {10-12}<10-1-1> twinning is activated in the ND specimen. The {10-12}<10-1-1> twinning is also marked by the white circle. As mentioned above, the optical microscopy data (Figure 3e,f) suggest that {10-12}<10-1-1> twinning occurs in the majority of globular α grains. It could be calculated from Figure 4e that the total volume fraction of {10-12}<10-1-1> twinning is 9.3% in the strain-limited ND specimen.

Figure 5a presents the OIM micrograph in the rectangle region of Figure 4e. The corresponding HCP unit cells of the globular α grains and the twinning are displayed in Figure 5b and Figure 5c, respectively. The {11-20} of the α grain is parallel to the normal of the RD–TD plane, which is in agreement with the initial texture of the alloy (Figure 2c). Table 2 summarizes the Schmidt factor (SF) of twinning and SF of slipping for two α grains in Figure 4e, in which {10-12}<10-1-1> twinning operates. Here we first attribute the formation of {10-12}<10-1-1> twinning to the appropriate SF of the α grains. It is reported that the formation of {10-12}<10-1-1> twinning is influenced by the resolved shear stress (RSS) of the grain [15,16,17]. The SF of {10-12}<10-1-1> twinning for grain 1 (G1) is 0.38, and the SF of {10-12}<10-1-1> twinning for grain 2 (G2) is 0.4. It is reported that the operation of {10-12}<10-1-1> twinning requires a proper SF for commercial-purity Ti and α Ti-alloy [18,19]. For example, Wang et al. pointed out that the SF for {10-12}<10-1-1> twinning of CP-Ti should be higher than 0.4 [20], and Yapici reported that the SF for {10-12}<10-1-1> twinning of TC4 alloy is 0.39 [21]. Our results are basically in accord with the previous works. Secondly, the SF of {10-12}<10-1-1> twinning is much higher than that of {0001}<11-20> slipping. The {0001}<11-20> basal slip is the most common deformation mechanism for the α phase of titanium alloys [4]. The SF of {10-12}<10-1-1> twinning is 4 times higher than the SF of {0001}<11-20> slipping in G1, and it is 10 times higher than the SF of {0001}<11-20> slipping in G2. These results indicate that the basal slip hardly operates in G1 and G2. Therefore, the suppression of basal slip may also lead to the formation of {10-12}<10-1-1> twinning.

#### 3.3.2. Dislocation Structures of Strain-Limited Specimen

Figure 6a,c,e show the TEM micrograph for the globular α grain of the strain-limited specimen that is loaded in the RD, TD and ND. Although the specimens are loaded in different directions, the dislocation structure of the globular α grain for the specimens is the same, which is the dislocation pile-up. In addition, the dislocation density is considerably low in the globular α grain. Figure 6b,d,f display the TEM micrographs for the fine α grain of the strain-limited specimen that is loaded in the RD, TD and ND. Although the compression directions are different, the dislocation structures of the fine α grain are all dislocation tangling. In addition, it should be pointed out that the dislocation density in the fine α grain of the strain-limited ND specimen is higher. It could also be seen from Figure 6 that the dislocation density of the globular α grain is much lower than that of the fine α grain. Combined with the twinning behaviors of the strain-limited specimen, we conclude that the initial dynamic compression of the RD specimen and the TD specimen is mainly dominated by the dislocation multiplication within the fine equiaxial α grains. However, the initial dynamic compression of the ND specimen is dominated by {10-12}<10-1-1> twinning of globular α grain and the dislocation multiplication within the fine equiaxial α grains.

### 3.4. Influence of Loading Direction on Adiabatic Shearing of TC4 Plate

Figure 7 shows the optical micrograph of the ruptured specimens. The ASB and the main crack could be found. The coarse α grains remain globular, and α laminates have been distorted drastically in the RD specimen and the TD specimen. In contrast, the dynamic compression of the ND specimen leads to the flattening of the coarse α grain, and the morphology of α laminate remains unchanged.

Figure 8 and the insertions are the optical micrographs recorded in the ASB region of the ruptured RD specimen. The vertical direction of the micrographs is the RD of the alloy plate. The cracks and two intersected ASBs that are marked as ASB1 and ASB2 could be found (50 μm scale bar). These results indicate that the fracture of the RD specimen is caused by the formation of ASBs. Insertion 1 is the high-magnification optical micrograph recorded in region 1. ASB1 and ASB2 are marked. The crack is located on the interface between ASB1 and the deformed matrix. Insertion 2 shows the high-magnification optical micrograph for region 2. The α phase exhibits the bright contrast, while β phase exhibits the dark contrast. Compared with the initial morphology of the β phase, we find that the β phase of the main ASB has transformed into nanoparticles because of the dramatic adiabatic shear. In addition, the original grain boundary of the coarse α grain disappears in the main ASB. Insertion 3 is a high-magnification micrograph for region 3, in which the main ASB1 is marked. It is interesting that another ASB, which is marked as ASB3, could be found in the vicinity of the crack.

Figure 9a shows the optical micrograph recorded in the ASB region of the ruptured TD specimen, and the high-magnification optical micrograph for the rectangle region of Figure 9a is displayed in Figure 9b. The vertical direction of these optical micrographs is the TD. Multiple ASBs are found, and ASB1 is the main ASB of the ruptured TD specimen. It could be seen from Figure 9b that the β phase is in the morphology of nanoparticles, which is the same as that of the ruptured RD specimen. These data suggest that the adiabatic shearing behavior of the TD specimen and the compressive fracture mechanism of the TD specimen are the same as those of the RD specimen.

Figure 10a shows the optical micrograph for the ASB region of the ruptured ND specimen, and Figure 10b displays the high-magnification optical micrograph for the rectangle region of Figure 10a. The vertical direction of these optical micrographs is the ND of the plate. Although the rupture of the ND specimen is also caused by the ASB, only one ASB could be found in the whole ruptured specimen. Within the ASB, the β phase is still in the nanoparticle morphology.

Figure 11a shows the TEM micrograph recorded in the center of the ASB for the RD specimen. The equiaxial nano-β grains could be found, which are marked by the circles. The TEM results on the morphology and the grain size of β grain are basically in good agreement with optical microscopy data of the β grain (Figure 8). These findings indicate that the dramatic shear of the ASB might lead to the refinement of the β phase and the dynamic recrystallization of β phase. Compared with the TEM micrograph for the α phase of the strain-limited specimen (Figure 6a,b), we find that the grain boundary of the α grain has disappeared, and the dislocation density for the α phase within the ASB increases considerably. In addition, the dislocation structure of the α phase has transformed into the dislocation cell, and a few nano-α laminates have formed within the dislocation cell and are marked by the rectangles. The finding of the nano-α laminate is consistent with the work of Zheng et al., in which the nano-α laminate was also found in the ASB of equiaxial TC4 alloy [5]. It is possible that the nano-α laminate evolves from the dislocation cell in the later period of the dramatic adiabatic shear.

Figure 11b shows the TEM micrograph recorded in the center of the ASB for the TD specimen. The equiaxial nano-β grains, the dislocation cell of the α phase and the nano-α laminate are marked by circles, arrows and rectangles, respectively. Generally speaking, the typical microstructures for the ASB of the TD specimen are basically the same as those of the RD specimen. Figure 11c shows the TEM micrograph recorded in the center of the ASB for the ND specimen. Although the equiaxial nano-β grains, the dislocation cell within the α phase and the nano-α laminate could be found, the volume fraction of the nano-α laminate within the dislocation cells is higher in the ASB. The formation of the abundant nano-α laminate suggests that the shearing plastic strain within the ASB of the ND specimen should be higher than that of the RD specimen and TD specimen.

## 4. Discussion

### 4.1. Dynamic Plasticity Anisotropy of TC4 Alloy Thick Plate

In the present work, the compressive dynamic properties of the mill-annealed TC4 alloy plate are evaluated in the RD, TD and ND. The loading direction hardly affects the dynamic yield strength or the dynamic flow stress of the plate. However, it strongly influences the dynamic plasticity of the plate. Since the dynamic compressive fracture mechanism of ND specimen, RD specimen and TD specimen is the shearing fracture induced by the ASB, the compressive fracture strain of the alloy (ε_f_) could be written as:ε_f_ = ε_E_ + ε_P_ + ε_ASB_
where ε_E_ is the elastic strain, ε_P_ is the plastic strain and ε_ASB_ is the strain caused by the ASB [22]. It could be seen from the strain–stress curves that ε_E_ is not strongly affected by the loading directions. In addition, ε_ASB_ is much lower than ε_P_. Therefore, the dynamic plasticity anisotropy of TC4 alloy thick plate is mainly caused by the anisotropy of ε_P_. Here we attribute the highest ε_P_ in ND of the plate to the texture of the coarse globular α grains and the distribution of α laminate.

First, owing to the {11-20} preferential orientation, most of the coarse globular α grains in the ND of the plate have the appropriate SF for {10-12}<10-1-1> twinning and are not appropriate for {0001}<11-20> slip [23,24]. As a result, the highest volume fraction of the {10-12}<10-1-1> twinning was activated in the ND specimen during the dynamic compression. The so-called twinning-induced plasticity (TWIP) has also been reported in titanium alloys, and its effect on improving the plasticity of titanium alloys is quite obvious [25]. In particular, a recent investigation indicates that the activation of {10-12}<10-1-1> twinning is favorable to the dynamic plasticity of TC4 alloy [26]. Therefore, the texture-related TWIP contributes to the highest ε_P_ in the ND of the plate.

Second, as the plate is loaded in the RD and TD, the normal of the α laminate is perpendicular to the loading direction. The special orientation of α laminate restricts the flattening of the coarse globular α grains and the fine equiaxial α grains. As a result, multiple strain localization zones form through the shearing of the α laminate and the severe distortion of the neighboring α grains in the late stage of uniform dynamic compression. These strain localization zones might rapidly transform into multiple ASBs because the strain localization zone is extremely close to the ASB. Therefore, owing to the strain localization, the uniform dynamic compression of the specimen tends to terminate at a relatively lower compression strain. In contrast, the normal of the α laminate is parallel to the loading direction as the plate is loaded in the ND. The straining of fine equiaxial α grains and the flattening of coarse globular α grains are not influenced by the α laminate. The local shearing of the α laminate does not occur, and the ND specimen is deformed continuously by the simultaneous straining of the equiaxial α grains and the coarse globular α grains. The uniform dynamic compression of the ND specimen terminates when it is dislocation-saturated. Therefore, the distribution of α laminate also contributes to the highest ε_P_ for the ND of the plate.

### 4.2. Non-Dynamic Recrystallization for the ASB of Mill-Annealed TC4 Alloy

Owing to the high shearing strain, it is common that the adiabatic shear could result in the dynamic recrystallization of the ASB for TC4 alloy [27,28]. For example, Eurr et al. proposed the ASB formation mechanism of equiaxial TC4 alloy, which includes dislocation multiplication induced by shearing, formation of the dense dislocation zone and DRX of α grain in the dense dislocation zone [27]. Similarly, Li et al. reported that the equiaxial nano-sized grains could be found in the ASB of pure-Ti, driven by the shear-rotational DRX mechanism [29]. However, in the present work, DRX of the α grain does not occur during the adiabatic shearing when the mill-annealed TC4 alloy plate is loaded in the RD, TD and ND. The recent work of Magagnosc et al. reveals that the onset of the DRX within the ASB of Ti-7Al titanium alloy is mainly driven by the accumulation of plastic strain [30]. As mentioned previously, when the plate is loaded in the RD and TD, multiple ASBs form. It is possible that the shearing strain in the ASB might be not high enough to generate the DRX of the α grain. Since the ASB of the ND specimen has more nano-sized α laminates, the shearing strain within the ASB of the ND specimen should be higher than that of the RD specimen and TD specimen. However, the shearing strain in the ASB of the ND specimen is still not high enough to result in the DRX of the α grain.

### 4.3. Delaying the ASB Formation of TC4 Alloy by Texture Design

The formation of the ASB and the fracture induced by the ASB have been frequently reported in the dynamic compression fracture of TC4 alloy. The ASB-formation strain of TC4 alloy is important for the impact resistance of components made of TC4 alloy because the ASB formation is accompanied by the initiation of the main crack. The well-known parameters that might affect the ASB-formation strain of TC4 alloy include the initial microstructure. For example, the work of Zheng et al. indicates that the ASB-formation strain of equiaxial TC4 alloy is higher than that of lamellar alloy bimodal alloy [31]. In the present work, we find a novel parameter that might improve the ASB-formation strain of TC4 alloy, which is the texture of the α phase. TC4 alloy exhibits various types of texture depending on the thermal–mechanical processing of the alloy. The results of the present work strongly indicate that the {11-20}-orientated α grain could delay the formation of the ASB in TC4 alloy through the preferential formation of {10-12}<10-1-1> twinning. Since the increase in the ASB-formation strain of TC4 alloy should improve the dynamic plasticity of TC4 alloy, the impact resistance of the TC4 alloy fan blade might be improved by texture optimization. The chord direction of the fan blade should be parallel to the ND of the mill-annealed TC4 alloy thick plate.

## 5. Conclusions

In this paper, the effects of the loading direction on dynamic mechanical property, deformation mechanism and adiabatic shearing behavior of a mill-annealed TC4 alloy plate have been investigated. The plasticity of the plate is anisotropic during dynamic compression. The fracture strain of the normal direction is 38.2% higher than that of the rolling direction and 32.2% higher than that of the traverse direction. The following conclusions were drawn:The mill-annealed TC4 alloy thick plate consists of globular α grain, fine equiaxial α grain, α laminate and β laminate. The normal of the α laminate is parallel to the normal direction of the plate. The α phase of the plate possesses {11-20} texture.The superior dynamic compressive plasticity in the ND of the TC4 plate is caused by {11-20} texture of the globular α grain, the distribution of the α laminate and the highest shear strain of the ASB.The {10-12}<10-1-1> twinning within the globular α grain, which is induced by {11-20}_α_ texture, might delay the ASB formation of mill-annealed TC4 alloy.Dynamic recrystallization of the α phase does not occur in the adiabatic shearing band of mill-annealed TC4 alloy.The strength and plasticity of the mill-annealed Ti-6Al-4V alloy with different dynamic compression strain levels will be investigated in the future.

## Figures and Tables

**Figure 1 materials-15-07047-f001:**
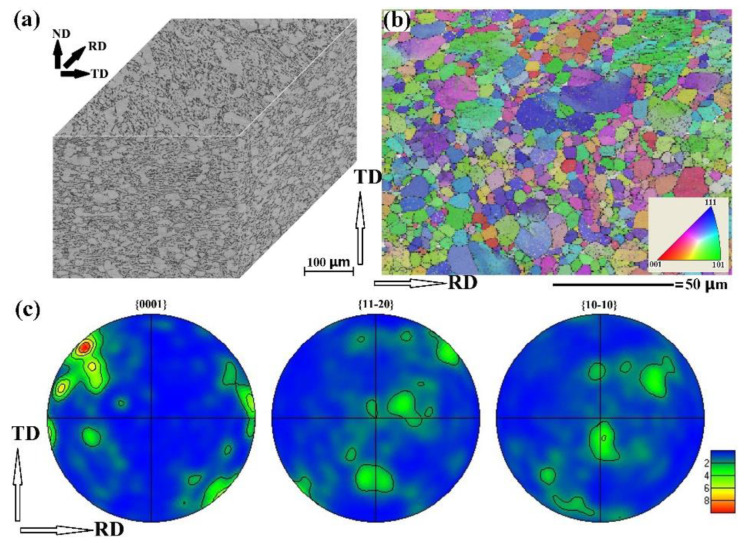
Initial microstructure of the mill-annealed TC4 alloy plate. (**a**) Optical micrographs recorded in the RD–TD plane, RD–ND plane and TD–ND plane of the plate; (**b**) orientation imaging microscopy (OIM) micrograph for α phase in the RD–TD plane; (**c**) α-{0001}, α-{11-20} and α-{10-10} pole figure of (**b**).

**Figure 2 materials-15-07047-f002:**
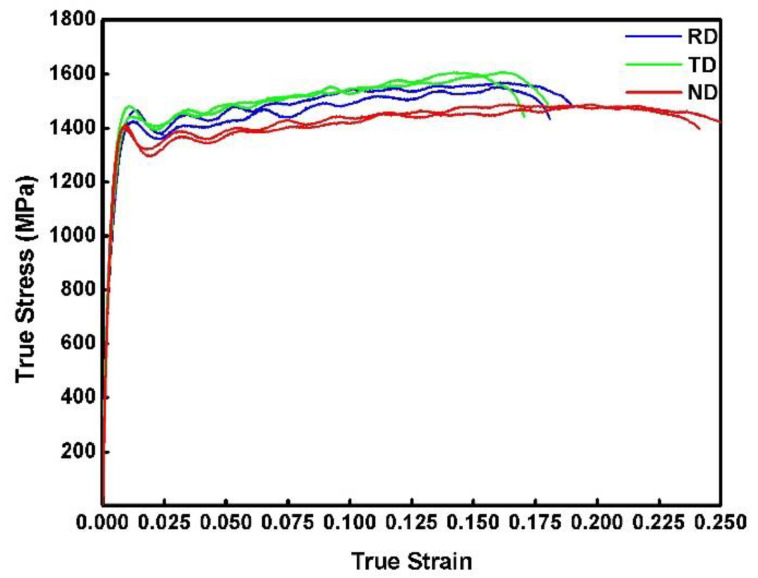
Dynamic compressive stress–strain curves for the mill-annealed Ti-6Al-4V alloy plate that is loaded in the RD, TD and ND.

**Figure 3 materials-15-07047-f003:**
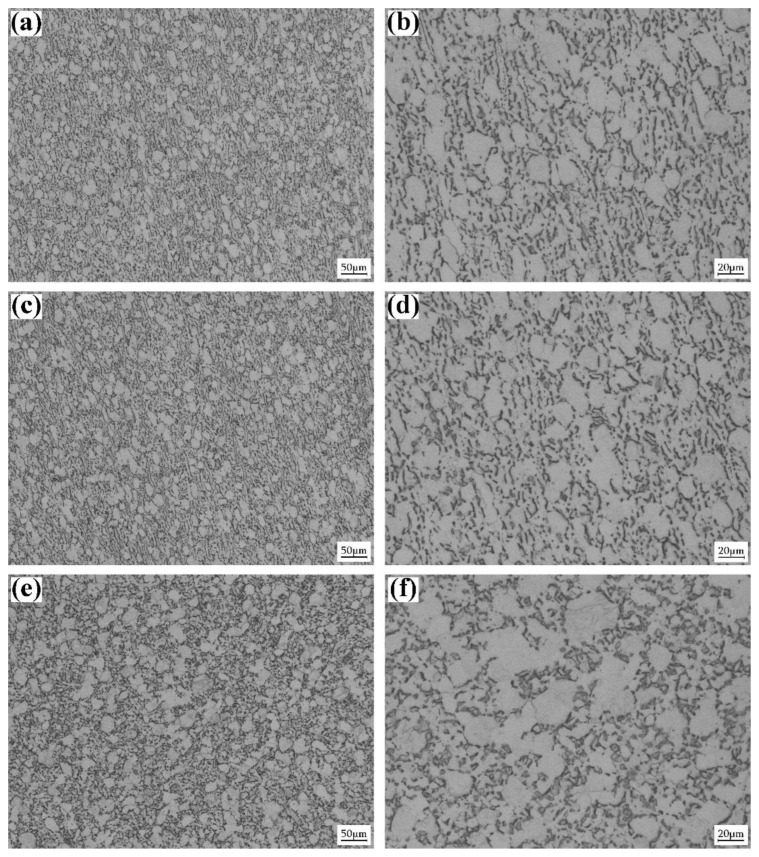
Optical micrographs of the mill-annealed Ti-6Al-4V alloy plate after the strain-limited (ε = 10%) dynamic compression test. (**a**,**b**) Strain-limited RD specimen; (**c**,**d**) strain-limited TD specimen; (**e**,**f**) strain-limited ND specimen.

**Figure 4 materials-15-07047-f004:**
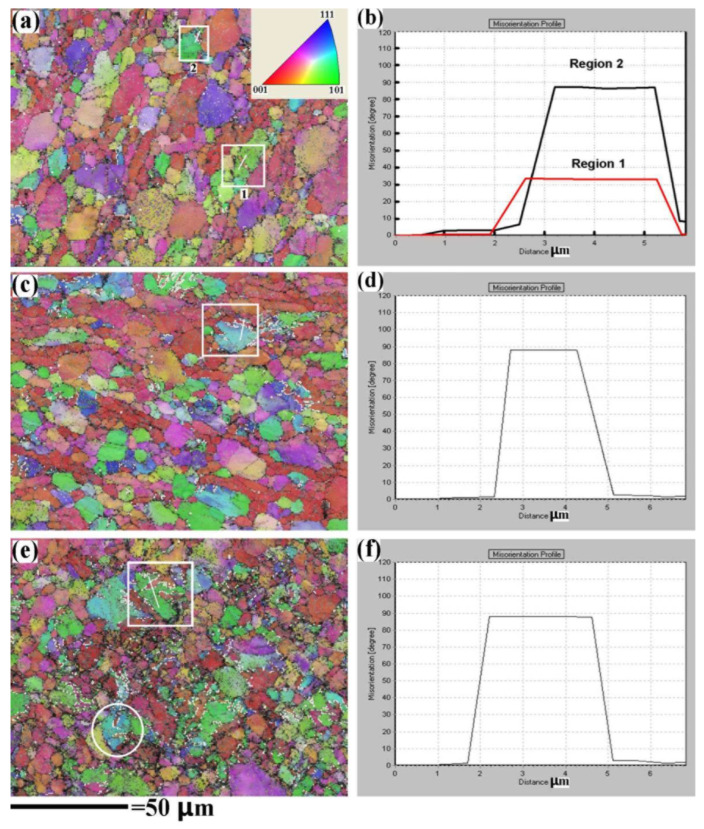
Twinning deformation behavior of the mill-annealed Ti-6Al-4V alloy plate after the strain-limited (ε = 10%) dynamic compression test. (**a**) OIM of the strain-limited RD specimen, RD is perpendicular to the micrograph. (**b**) Misorientation profile of the line in region 1 and region 2 in (**a**). (**c**) OIM of the strain-limited TD specimen, TD is perpendicular to the micrograph. (**d**) Misorientation profile of the line in the region of (**c**). (**e**) OIM of the strain-limited ND specimen, ND is perpendicular to the micrograph. (**f**) Misorientation profile of the line in the region of (**e**).

**Figure 5 materials-15-07047-f005:**
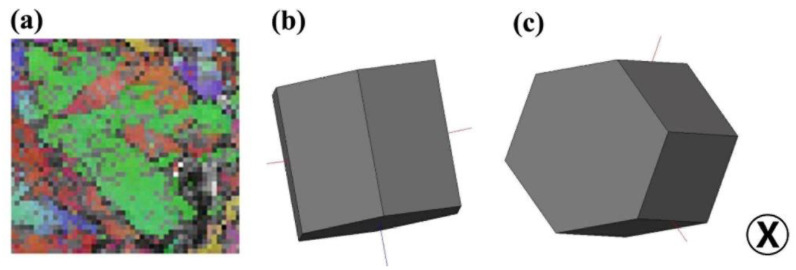
OIM for {10-12}<10-1-1> twinning in one globular α grain. (**a**) OIM in the rectangle region of Figure 4e. (**b**) Orientation of unit cell for matrix (green region in (**a**)). (**c**) Orientation of unit cell for twinning (red region in (**a**)), the “X” stands for the normal direction of the plate.

**Figure 6 materials-15-07047-f006:**
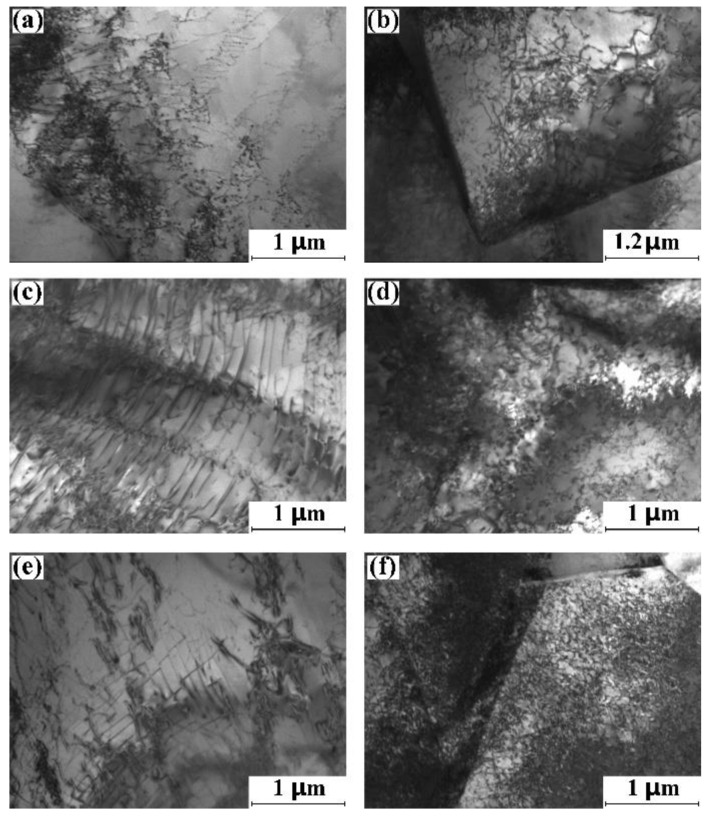
TEM micrograph of the dislocation structure of the mill-annealed Ti-6Al-4V alloy plate after the strain-limited (ε = 10%) dynamic compression test. The loading direction is parallel to the normal direction of the TEM micrograph. (**a**) Dislocation pile-up for the globular α grain of RD specimen, (**b**) dislocation tangling for the fine equiaxial α grain of RD specimen, (**c**) dislocation pile-up for the globular α grain of TD specimen, (**d**) dislocation tangling for the fine equiaxial α grain of TD specimen, (**e**) dislocation pile-up for the globular α grain of ND specimen, (**f**) dislocation tangling for the fine equiaxial α grain of ND specimen.

**Figure 7 materials-15-07047-f007:**
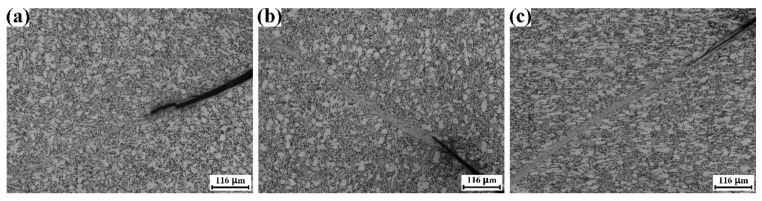
Optical micrographs of the mill-annealed Ti-6Al-4V alloy plate after the dynamic compression test. The loading direction is vertical. (**a**) Ruptured RD specimen, (**b**) ruptured TD specimen, (**c**) ruptured ND specimen.

**Figure 8 materials-15-07047-f008:**
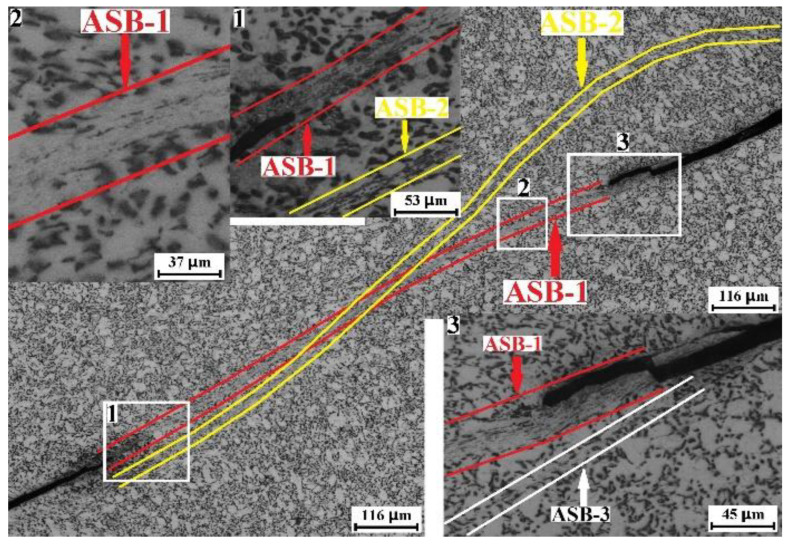
Optical micrographs regarding the adiabatic shearing behavior of mill-annealed Ti-6Al-4V alloy plate compressed in the RD. The loading direction is vertical. Insertion 1, insertion 2 and insertion 3 are the high-magnification optical micrographs for the rectangle region 1, rectangle region 2 and rectangle region 3.

**Figure 9 materials-15-07047-f009:**
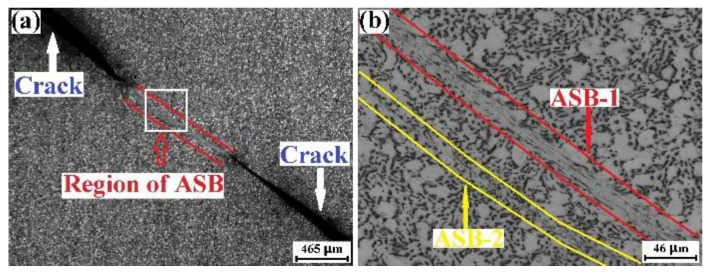
Optical micrographs regarding the adiabatic shearing behavior of mill-annealed Ti-6Al-4V alloy plate compressed in the TD. The loading direction is vertical. (**a**) Optical micrograph for the ASB and the crack induced by the ASB, (**b**) high-magnification optical micrograph for the rectangle region of (**a**).

**Figure 10 materials-15-07047-f010:**
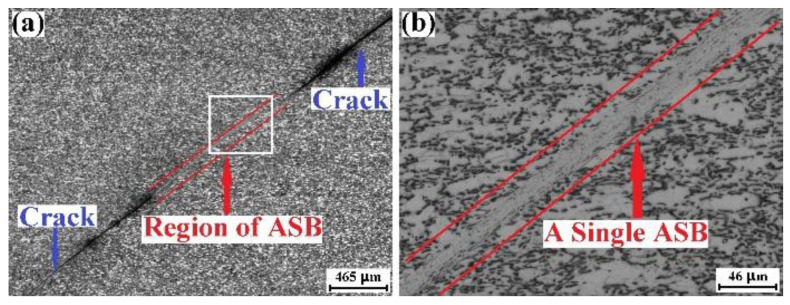
Optical micrographs regarding the adiabatic shearing behavior of mill-annealed Ti-6Al-4V alloy plate compressed in the ND. The loading direction is vertical. (**a**) Optical micrograph for the ASB and the crack induced by the ASB, (**b**) high-magnification optical micrograph for the rectangle region of (**a**).

**Figure 11 materials-15-07047-f011:**
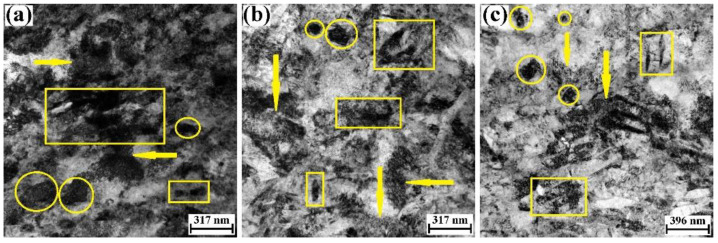
Microstructure of ASB for mill-annealed Ti-6Al-4V alloy plate. (**a**) TEM micrograph for ASB of RD specimen, (**b**) TEM micrograph for ASB of TD specimen, (**c**) TEM micrograph for ASB of ND specimen. The equiaxial nano-β grains, the dislocation cell of the α phase and the nano-α laminate are marked by circles, arrows and rectangles.

**Table 1 materials-15-07047-t001:** Dynamic compressive mechanical performance of TC4 alloy plate.

Loading Directions	Dynamic Yield Strength (MPa)	Dynamic Fracture Strain
RD	1427 ± 30	0.175 ± 0.017
TD	1454 ± 18	0.183 ± 0.015
ND	1408 ± 9	0.242 ± 0.009

**Table 2 materials-15-07047-t002:** Schmidt factor of twinning and slipping for the grains containing {10-12} <10-1-1> twinning.

Deformation Systems	Grain 1	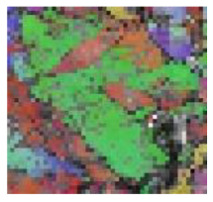	Grain 2	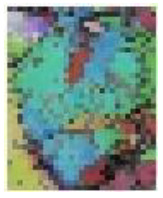
**Twinning**: {10-12}<10-1-1>	0.38		0.4	
**Slipping**: {0001}<11-20>	0.09		0.04	

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
