# Peer review of "Influence of Loading Directions on Dynamic Compressive Properties of Mill-Annealed Ti-6Al-4V Thick Plate"

_materials, 2022, doi:10.3390/ma15207047_

Round 1

Reviewer 1 Report

Dear Editor

The current work deals with the effect of loading direction on the dynamic flow stress, dynamic plasticity and fracture behavior of Ti64 alloy. This is a comprehensive work which has been very well planned and executed and would be interesting to readers. I recommend this paper for publication, after considering the following minor comments:

Comment 1: In line 250 the sentence “In addition, it should be pointed out that the dislocation density of the strain-limited ND specimen is lower” is correct only about the globular α grain in Figure 6 (a, c and e) but in the case of fine α grain the dislocation density is expected to be higher, in Figure 6 (b, d and f).

Comment 2: The presence of the equiaxed nano-β grains and the nano-α laminate how is identified in Fig 11?

Comment 3: Line-scale in Fig 8 is very illegible, also grain size of β grains in this fig (the insertion 2) is not in well agreement with TEM result in Fig. 11 (In contrast with line 326-327 manuscript)

Comment 4: This article well discussed about the relation between texture and activation of twinning induced plasticity in Ti-6Al-4V alloy. This also applies about tensile deformation behavior of Ti-6Al-4V alloy that you can see in  https://doi.org/10.1016/j.jmrt.2020.12.110. You may have some comparative discussion with https://doi.org/10.1016/j.jmrt.2022.02.105

Sincerely yours

Author Response

1.In line 250 the sentence “In addition, it should be pointed out that the dislocation density of the strain-limited ND specimen is lower” is correct only about the globular α grain in Figure 6 (a, c and e) but in the case of fine α grain the dislocation density is expected to be higher, in Figure 6 (b, d and f).

The sentence “In addition, it should be pointed out that the dislocation density of the strain-limited ND specimen is lower” has been revised as “In addition, it should be pointed out that the dislocation density in the fine α grain of the strain-limited ND specimen is higher”.

  1. The presence of the equiaxednano-β grains and the nano-α laminate how is identified in Fig 11?

Our experimental dataon the chemical composition of the adiabatic shearing band suggest that V is enriched in the nano-equiaxed grain and Al is enriched in the nano-laminate. Consequently, the nano-equiaxed grain could be identified as Beta phase, while the nano-laminate could beidentified as Alpha phase.

  1. Line-scale in Fig 8 is very illegible, also grain size of β grains in this fig (the insertion 2) is not in well agreement with TEM result in Fig. 11 (In contrast with line 326-327 manuscript).

The line-scale in figure 8 has been enlarged.

In fact, two kinds of Beta phase are could be observed in figure 8. The first kind of Beta phase is in the transition region of the adiabatic shearing band, and it is much coarser than the Beta phase in figure 11. In contrast, the second kind of Beta phase is inside the adiabatic shearing band. Owing to the dramatic shearing in the adiabatic shearing band of the alloy,the second kind of Beta phase is much finer than the first kind of the Beta phase. The grain size for the second kind of Beta grains is close to the TEM result in figure 11.

  1. This article well discussed about the relation between texture and activation of twinning induced plasticity in Ti-6Al-4V alloy. This also applies about tensile deformation behavior of Ti-6Al-4V alloy that you can see inhttps://doi.org/10.1016/j.jmrt.2020.12.110. You may have some comparative discussion withhttps://doi.org/10.1016/j.jmrt.2022.02.105.

We have read the paper entitled “Compressive/tensile deformation behavior and the correlated microstructure evolution of Ti-6Al-4V titanium alloy at warm temperatures” and the paper entitled “The correlation of c-to-a axial ratio and slip activity of martensite including microstructures during thermomechanical processing of Ti-6Al-4V alloy”. We have cited these papers, because these papers are considerable helpful in understanding the competition between the slip and twinning in the Alpha phaseof Ti-6Al-4V titanium alloy, which is found in the present work.

Reviewer 2 Report

The authors present the results of research on the influence of loading directions on dynamic compressive properties of mill-annealed Ti-6Al-4V thick plate.

Important notes on the manuscript:

1.       The literature review has been prepared correctly, but the description of the purpose of undertaking research and possible benefits of the obtained results in relation to the currently known achievements in this field needs to be extended.

2.       Could the authors in chapter 2 "Material and methods" extend the description of the tested material with its mechanical properties? (for the tested titanium alloy).

3.       The authors described the microstructure of the alloy after dynamic deformation in great detail, but the description itself is more of an observation rather than a scientific consideration. In my opinion, it would be good to extend this description to a cause-and-effect analysis.

4.       Have there been any strength tests on the deformed alloy?

Author Response

  1. The literature review has been prepared correctly, but the description of the purpose of undertaking research and possible benefits of the obtained results in relation to the currently known achievements in this field needs to be extended.

The description of the aim of the present research and the possible benefits of the findings has been added to the introduction section.

  1. The authors described the microstructure of the alloy after dynamic deformation in great detail, but the description itself is more of an observation rather than a scientific consideration. In my opinion, it would be good to extend this description to a cause-and-effect analysis.

The aim of the present work is to explore the dynamic mechanical properties of the mill-annealed Ti-6Al-4V titanium alloy thick plate that is used to fabricate the fan blade of the compressor. We originally assumed that the yield strength and the fracture strain of the thick plate would be isotropic during highstrain rate compression. However, these results contradict the original hypothesis: The compressive stress-strain curves of the plate suggest that the plasticity of the thick plate is anisotropic.In this paper, we have made every effort to present the scientific interpretation of the dynamic plasticityanisotropy for the thick plate, which is supported by the microstructure observation.

  1. Have there been any strength tests on the deformed alloy? Could the authors in chapter 2 "Material and methods" extend the description of the tested material with its mechanical properties?(for the tested titanium alloy).

The work regarding the strength of the Ti-6Al-4V titanium alloy after the high-strain rate compression should be a good topic and is of great importance to Ti-6Al-4V compressor blade. However, until now we have not conduct the strength tests on the deformed alloy. We plan to investigate the strength of deformed alloy in further by measuring the hardness, the quasi-static yield strength and the dynamicyield strength of the strain-limitedspecimen with different strain levels, which is based on the present work and has been add to the conclusion section of the paper. We would like to continue to submit our further work to this journaland sincerely hope that the reviewer 2 could help us to examine our further work.

Reviewer 3 Report

In this manuscript the authors investigate the effect of the loading direction on the behavior of mill-annealed Ti-6Al-4V alloy plate under dynamic compressive loading. This is an interesting manuscript. The authors here report the results of compressive experiments and characterization such as optical microscopy, SEM and TEM. The results seem to be sound and useful, therefore, I suggest this manuscript for publication once the authors make changes according to my comments below:

1.      In the introduction the authors mentioned that they load the specimens at strain rates of 1000/s; however, in Section 2.2 and Section 2.3 the authors state that the strain rate was 2000/s. So what was the actual strain rate? Please, clarify as strain rate is a very important parameter and often affects compressive strength and behavior significantly.

2.      Line 18: “The of activation {10-12}<10-1-1> twinning …”. Please, correct this typo.

3.      Line 50: “… are inversely proportional to …”; is -> are.

4.      Lines 55-57: I think the reference is missing in this sentence.

5.      Line 58: “radial direction of the rod”. “of” is missing.

6.      Line 78: “Dynamic compression specimen”. I think it is better to say “Specimen for dynamic compression”.

7.      Line 99: I suggest adding a comma after the word “test” or even rephrasing this sentence into something like “Before starting dynamic compression test, …”. In addition, replace “on the end” with “at the end”.

8.      Lines 158-159: “The dynamic yielding strength of ND specimen is only 1.3%”. Something is wrong here. Perhaps, the authors meant yield strain? If the authors meant strength, then they should provide a number in MPa. In addition, I think it is correct to say “yield strength/strain” and not “yielding strength/strain”. I suggest that the authors make these corrections throughout the manuscript.

9.      Lines 161-162: “suggesting that plate is plasticity anisotropy in dynamic compression”. I suggest that the authors correct this sentence as “suggesting that the plate is plastically anisotropic …”.

10.  Line 174: I suggest that the authors replace the word ”obviously” with “evidently”.

11.  Figure 4: It is hard to see what’s on the axes in figures 4b,d,f. I suggest that the authors increase the font size.

12.  Figures 7-10: It is almost impossible to see scale bars in these figures. I suggest that the authors correct this.

13.  Lines 359-360: “… because that εE are not strongly affected by the loading directions”. Please correct this sentence since the point is confusing. Did the authors mean “… because that of εE is not …”?

14.  Line 422: “dynamic compression property” sounds weird. Since this is one property, why not state this property?

Author Response

  1. In the introduction the authors mentioned that they load the specimens at strain rates of 1000/s; however, in Section 2.2 and Section 2.3 the authors state that the strain rate was 2000/s. So what was the actual strain rate? Please, clarify as strain rate is a very important parameter and often affects compressive strength and behavior significantly.

The strain rate of the dynamic compression test is 2000/s. The strain rate mentioned in the introduction has been revised.

  1. Line 18: “The of activation {10-12}<10-1-1> twinning …”. Please, correct this typo.

In line 18, the sentence “The of activation {10-12}<10-1-1> twinningcould delayformation of the adiabatic shearing band(ASB)” has been revised as “The activation of {10-12}<10-1-1> twinningcould delayformation of the adiabatic shearing band(ASB)”.

  1. Line 50: “…areinverselyproportional to …”; is -> are.

In line 50, the sentence “Yield strength and flow stress of TC4 alloy is inverse proportional to deformation temperature during high strain rate deformation” has been revised as “Yield strength and flow stress of TC4 alloy are inverse proportional to deformation temperature during high strain rate deformation”.

  1. Lines 55-57: I think the reference is missing in this sentence.

The reference 12 has been added to the sentence “Gu et al. found that the equiaxialTi-64 alloy rod exhibits obviousplasticityanisotropy in during the dynamic compression [12]”.

  1. Line 58: “radial direction of the rod”. “of” is missing.

In line 58, “of” has been added to the sentence “while the fracture strain in the radial direction of the rod is only 0.17”

  1. Line 78: “Dynamic compression specimen”. I think it is better to say “Specimen for dynamic compression”.

In line 78, the “Dynamic compression specimen” has been revised as “specimen for dynamic compression”.

  1. Line 99: I suggest adding a comma after the word “test” or even rephrasing this sentence into something like “Before starting dynamic compression test, …”. In addition, replace “on the end” with “at the end”.

The sentence “Before dynamic compression test the stopping ring was fixed on the end of the transmitted bar by 3M-DP460 epoxy adhesive” has been revised as “Before starting the dynamic compression test, the stopping ring was fixed at the end of the transmitted bar by 3M-DP460 epoxy adhesive.”

  1. Lines 158-159: “The dynamic yielding strength of ND specimen is only 1.3%”. Something is wrong here. Perhaps, the authors meant yield strain? If the authors meant strength, then they should provide a number in MPa. In addition, I think it is correct to say “yield strength/strain” and not “yielding strength/strain”. I suggest that the authors make these corrections throughout the manuscript.

The sentence “The dynamic yielding strength of ND specimen is only 1.3% and 3.1% lower than that of RD and TD specimen” has been revised as “The dynamic yield strength of ND specimen is 1.3 percent lower than that of RD specimen and is 3.1 percent lower than that of TD specimen”.

9.Lines 161-162: “suggesting that plate is plasticity anisotropy in dynamic compression”. I suggest that the authors correct this sentence as “suggesting that the plate is plastically anisotropic …”.

The sentence “suggesting that plate is plasticity anisotropy in dynamic compression” has been revised as “suggesting the plate is plastically anisotropic in dynamic compression”.

10.Line 174: I suggest that the authors replace the word ”obviously” with “evidently”.

In line 174, the word “obviously” has been revised as “evidently”.

11.Figure 4: It is hard to see what’s on the axes in figures 4b,d,f. I suggest that the authors increase the font size.

We have increased the font size of figure 4(b), figure 4(d) and figure 4(f).

12.Figures 7-10: It is almost impossible to see scale bars in these figures. I suggest that the authors correct this.

In the revised paper, we have revised the scale bars for these figures.

13.Lines 359-360: “… because that εE are not strongly affected by the loading directions”. Please correct this sentence since the point is confusing. Did the authors mean “… because that of εE is not …”?

In the revised manuscript, we have revised the discussion as “It could be seen from the strain-stress curves that εE are not strongly affected by the loading directions. In addition, εASB is much lower than εP. Therefore, thedynamic plasticity anisotropy of TC4 alloy thick plate is mainly caused by the anisotropy ofεP.”.

14.Line 422: “dynamic compression property” sounds weird. Since this is one property, why not state this property?

In line 422, the sentence “In this paper, effects of the loading direction on the dynamic compression property, the deformation mechanism and the adiabatic shearing behavior of mill-annealed TC4 alloy plate have been investigated” has been revised as “In this paper, effects of the loading direction on dynamicmechanical property, deformation mechanism and adiabatic shearing behavior of mill-annealed TC4 alloy plate have been investigated”.